# New Aspect of Composition and Biological Properties of *Glechoma hederacea* L. Herb: Detailed Phytochemical Analysis and Evaluation of Antioxidant, Anticoagulant Activity and Toxicity in Selected Human Cells and Plasma In Vitro

**DOI:** 10.3390/nu15071671

**Published:** 2023-03-29

**Authors:** Natalia Sławińska, Magdalena Kluska, Barbara Moniuszko-Szajwaj, Anna Stochmal, Katarzyna Woźniak, Beata Olas

**Affiliations:** 1Department of General Biochemistry, Faculty of Biology and Environmental Protection, University of Lodz, 90-236 Łódź, Poland; 2Department of Molecular Genetics, Faculty of Biology and Environmental Protection, University of Lodz, 90-236 Łódź, Poland; 3Department of Biochemistry and Crop Quality, Institute of Soil Science and Plant Cultivation, State Research Institute, 24-100 Puławy, Poland

**Keywords:** cell viability, DNA damage, *Glechoma hederacea* L., hemostasis, oxidative stress, peripheral blood mononuclear cells

## Abstract

It is known that phenolic compounds can alleviate the negative impact of oxidative stress and modulate hemostasis. However, the effect of extracts and phenolics from *Glechoma hederacea* L. on the biomarkers of these processes is not well documented. The aim of our study was to investigate the in vitro protective effects of one extract and three fractions (20, 60, and 85% fraction) from *G. hederacea* L. on oxidative stress and hemostasis. Phytochemical analysis showed that aerial parts of *G. hederacea* L. are rich in both phenolic acids (such as rosmarinic acid, neochlorogenic acid, and chlorogenic acid) and flavonoids (mainly rutin and glycoside derivatives of apigenin, quercetin, and luteolin). We observed that the 85% fraction (at three concentrations: 5, 10, and 50 μg/mL) inhibited protein carbonylation. Moreover, the extract and 85% fraction (at the concentration of 50 μg/mL) could reduce lipid peroxidation. All fractions and the extract were very effective at decreasing H_2_O_2_-induced DNA damage in PBM cells. The 85% fraction had the strongest protective potential against DNA oxidative damage. We also observed that the extract and fractions decreased PBM cell viability to a maximum of 65% after 24 h incubation. Our results indicate that the 85% fraction showed the strongest antioxidant potential. The main component of the 85% fraction was apigenin (26.17 ± 1.44 mg/g), which is most likely responsible for its strong antioxidant properties.

## 1. Introduction

*Glechoma hederacea* L. (Lamiaceae) is a plant belonging to the *Labiatae* family, which is commonly known as “ground ivy” and “gill over the ground”. It is widely distributed in China, Korea, and Japan. In addition, the species is widespread in almost all of Europe, the Caucasus and Siberia, as well as in North America. Numerous studies have shown that it possesses various beneficial effects. In traditional Chinese medicine, this plant is prescribed to patients with diabetes, cholelithiasis, inflammation, dropsy, and abscess [1]. It also has a positive effect on immune, respiratory, and urinary systems, improves the condition of skin and hair, and has bactericidal and fungicidal effects [1,2,3,4,5]. 

Phenolic compounds are its main bioactive components [1,2,3,4,5]. Ground ivy is also rich in essential oils, vitamin C, provitamin A, zinc, iron, silicon, molybdenum, and calcium [1,2,3,4,5]. 

*G. hederacea* has a strong smell and a bitter, spicy taste. Because of these properties, it is a commercially available food product. Gill tea, which was used in England since the 18th century, can be brought up as an example. Dried aerial parts of *G. hederacea* are gaining popularity as a spice. Moreover, fresh, chopped herbs can be added to scrambled eggs, omelettes, herbal butter, and cheese. It also enriches the taste of potato salads, boiled potatoes, rice, or pasta. In addition, breweries sometimes use this plant as a source of bitterness in beer production [5].

Oxidative stress is considered a common pathologic mechanism of different diseases, including cardiovascular diseases and cancer. It is usually attributed to the excessive production of reactive oxygen species (ROS), which leads to DNA damage, protein carbonylation, and lipid peroxidation. In addition, oxidative stress can also modulate hemostasis [6]. 

In recent years, plant-derived compounds have demonstrated numerous biological activities. Antioxidants present in plants play an important role in maintaining human health. Phytochemicals sometimes have anticoagulant or procoagulant properties as well. It is known that some compounds isolated from plants (especially polyphenols) can mitigate the negative impact of oxidative stress and regulate hemostasis [7,8]. 

Milovanovic et al. [9] studied the antioxidant potential of *G. hederacea* as a food additive. The ethanol–water (8:2, *v/v*) and purified ethyl acetate extracts had significantly stronger antioxidant properties than other used extracts and commercial antioxidants such as tocopherol and butylohydroxyanizol mixture. The results of Chou et al. [2] also showed the in vitro antioxidant potential of a hot water extract of *G. hederacea* (100–400 μg/mL), in which chlorogenic acid, rosmarinic acid, caffeic acid, genistein, rutin, and ferulic acid were the most abundant phytochemicals. It prevented LPS-induced DNA damage in RAW264.7 macrophages, decreased the level of malondialdehyde, increased the concentration of glutathione, and regulated the activity of antioxidant enzymes (catalase, glutathione peroxidase, and superoxide dismutase).

However, the effect of extracts and phenolics from *G. hederacea* on biomarkers of these processes is not well documented. The aim of our study was to investigate the protective effects of one extract and three fractions (20, 60, and 85% fraction) from *G. hederacea* on oxidative stress and hemostasis in vitro. We measured the levels of two biomarkers of oxidative stress in human plasma treated with H_2_O_2_/Fe^2+^: protein carbonylation and lipid peroxidation. Moreover, we studied the effect of this extract and these fractions on the viability of peripheral blood mononuclear (PBM) cells and the level of DNA oxidative damage induced by hydrogen peroxide. We also measured their effects on three hemostatic parameters of human plasma: prothrombin time (PT), thrombin time (TT), and activated partial thromboplastin time (APTT). The concentration of plant extract and fractions (≤50 μg/mL) used in our study can be obtained through oral administration, which is an important consideration for practical applications [10].

## 2. Materials and Methods

### 2.1. Reagents

Phosphate-buffered saline (PBS), low-melting-point (LMP) and normal-melting-point (NMP) agarose, 4′,6-diamidino-2-phenylindole (DAPI), dimethyl sulfoxide (DMSO), resazurin sodium salt, thiobarbituric acid (TBA), and hydrogen peroxide (H_2_O_2_) were purchased from Sigma-Aldrich. Trichloroacetic acid (TCA) and NaCl was purchased from POCH (Avantor performance materials, Gliwice, Poland). Reagents needed for coagulation time measurements were purchased from Kselmed (Grudziądz, Poland). Other reagents were purchased from commercial distributors and were of the highest available grade. 

### 2.2. Plant Material

The aerial parts of *Glechoma hederacea* were collected from a wild site located in the village of Łęka, Lubelskie Voivodeship, Poland (21°54 N, 51°270 E). The plants were harvested after flowering (from a well-lit place) in May 2022 and frozen. Frozen samples were lyophilized (CHRIST Gamma 2-293 16 LSC Freeze Dryers, Osterode am Harz, Germany). A voucher specimen (IUNG/GH/2021/1) was deposited at the Department of Biochemistry and Crop Quality, Institute of Soil Science and Plant Cultivation, State Research Institute, Puławy, Poland.

### 2.3. Preparation of the Extract and Fractions from Aerial Parts of G. hederacea

The freeze-dried aerial parts of *G. hederacea* were milled in a laboratory mill (ZM200, Retsch, Haan, Germany) and then sieved through a 0.5 mm sieve. The obtained powder was extracted with 70% methanol (*v*/*v*) in a ratio of 1:20 during 5 h. The extraction was supported by sonication in an ultrasonic bath (room temperature, 15 min per each extraction hour). The content was centrifuged at 4000×ss *g* for 10 min. The residue was extracted twice under the same conditions as above and centrifuged. The supernatants were pooled together and concentrated, which was carried out under reduced pressure. 

Then, the extract was transferred to a preconditioned RP-C18 column (65 × 30 mm, 140 μm; Cosmosil C18-PREP; Nacalai Tesque, Kyoto, Japan). The volume of the extract loaded on the column was 5%. Polar compounds were removed with 1% methanol and 0.1% formic acid, *v*/*v*), and active metabolites were eluted with 85% methanol with 0.1% formic acid (*v*/*v*) to give a purified extract, which was divided into two parts. One part was used as research material, and the second part was applied to the same RP-C18 column, and the active metabolites were fractionated by eluting with 20 methanol (*v*/*v*) to obtain the 20% fraction, 60% methanol (*v*/*v*) to obtain the 60% fraction, and the rest of the compounds were eluted with 85% methanol (*v*/*v*) to obtain the 85% fraction. The elution volume for each fraction was 250 mL. All fractions were freeze-dried. Fraction 20% constituted 36.6%, fraction 60–58.1% and fraction 85–5.3% of this separation. Finally, 3 mg of extract and fractions were dissolved in 1 mL of 70% methanol. Then, 5 μL of each sample was subjected to qualitative analysis with UHPLC-QTOF-MS, while 3 μL was used for UHPLC-MS analysis to determine the concentration of phenolic acids and flavonoids.

### 2.4. The Qualitative Analysis Using Ultra-High-Resolution Mass Spectrometry UHPLC-QTOF-MS

The qualitative investigations of the extract and three fractions (20%, 60% and 85%) were performed according to previously described procedures in Rolnik et al. [11]. They were determined by high-resolution LC-MS (HR-ESI-MS) analyses which were performed with the Thermo Ultimate 3000 RS (Thermo Fischer Scientific, Waltham, MA, USA) chromatographic system coupled with a Bruker Impact II HD (Bruker, Billerica, MA, USA) quadrupole-time of flight (Q-TOF) mass spectrometer and CAD detector (Charged Aerosol Detector).

The chromatographic separation was carried out on a Waters HSS T3 column (150 × 2.1 mm, 1.8 μm, Wexford, Ireland) at 40 °C, and the flow rate was 400 μL/min. A linear gradient used to separate analytes was as follows: from 2% acetonitrile in 0.1% formic acid to 99% acetonitrile in 0.1% formic acid over 22 min. The sample injection volume was 5.0 μL.

The compounds were analyzed based on data from UV and mass spectra. Electrospray ionization (ESI) was performed in negative and positive ion mode. The mass scan range was set from 80 to 2000 *m*/*z*. Ions source parameters: capillary voltage 3.0 kV, dry gas 6.0 L/min and dry temperature 200 °C. The PDA was operated in the range of 190–750 nm. Data processing was performed using DataAnalysis 4.3 (Bruker Daltonik GmbH, Bremen, Germany). 

### 2.5. Ultra-High-Pressure Liquid Chromatography (UHPLC-MS) Conditions 

The quantitation of flavonoids and phenolic acids was performed with an ACQUITY UPLC system, which was equipped with a triple quadrupole mass detector and a PDA (TQD, Waters, Milford, MA, USA). The separation of compounds was carried out with an Acquity UPLC BEH C18 column (100 × 2.1 mm, 1.7 μm particle size; Waters, Wexford, Ireland) with a gradient mobile phase. Solvent A—0.1% formic acid and solvent B—acetonitrile with 0.1% formic acid were used as follows: 6–33% of B in 10.9 min at a flow rate of 400 μL/min. The temperature of the column was maintained at 45 °C. The injection volume was 3 μL. Compound identification was carried out on the basis of data from mass spectra. The ESI ionization was carried out in negative ion mode. The PDA was operated in the range of 191–480 nm; the resolution was 3.6 nm. Obtained data were processed with MassLynx V4.1 software, Waters. The quantitative analysis of the analyzed compounds was carried out with the help of data from UV spectra (350 nm for flavonoids and 320 nm for phenolic acids). The quantity of compounds was determined with an external standard method. The results were expressed as mg/g of extract and fractions. The linearity of the method was shown with a calibration curve, which used eight known concentrations of the standard (0.1–325 μg/mL). The linear correlation coefficient (R^2^) for the curve of chlorogenic acid and rutin was 0.9998 for phenolic acids and 0.999 for flavonoids respectively. 

### 2.6. Stock Solution of Extract and Fractions from Aerial Parts of G. hederacea 

To make stock solutions, 10 mg of lyophilized extract or fractions was dissolved in 1:1 (*v/v*) distilled water/DMSO. Then, stock solutions were diluted to obtain working solutions at concentrations of 100, 500, 1000 and 5000 μg/mL. The final concentrations of the extract or fractions in the biological samples were 1, 5, 10, and 50 μg/mL.

### 2.7. Plasma Isolation 

Human whole blood was collected from healthy (*n* = 6, aged 25–28) medication-free and non-smoking donors (male (*n* = 3) and female (*n* = 3)). The blood was drawn at the “Diagnostyka” blood collection center on Brzechwy 7a St in Lodz, Poland. The volunteers did not take any addictive substances (e.g., tobacco, alcohol), antioxidant supplementation, or any other substances that could influence oxidative status or hemostasis. The study was accepted by the Committee for Research on Human Subjects of the University of Lodz (the number of permission is 3/KBBN-UŁ/II/2016). Blood was drawn into CPDA1 (Citrate, Phosphate, Dextrose, Adenine) tubes. Plasma was obtained by differential centrifugation (2800× *g*, 20 min).

### 2.8. Coagulation Times of Human Plasma

Coagulation times were determined coagulometrically with an Optic Coagulation Analyser, model K-3002 (Kselmed, Grudziadz, Poland), according to the method described by Malinowska et al. [12]. The reagents were purchased from Kselmed. The samples containing human plasma and the extract or fractions (at concentrations: 1, 5, 10, and 50 μg/mL) were incubated at 37 °C for 30 min. In control, 0.9% NaCl was used instead of the extract and fractions. To measure the prothrombin time, 50 μL of the samples was incubated again at 37 °C for 2 min, and 100 μL of Dia-PT solution was added immediately before the measurements. To measure the thrombin time, 50 μL of the samples was incubated at 37 °C for 1 min, and 100 μL of thrombin (at the final concentration of 5 U/mL) was added immediately before the start of the measurement. To measure the activated partial thromboplastin time, 50 μL of the samples was incubated at 37 °C for 3 min with 50 μL of Dia-PTT solution; after the incubation, 50 μL of Dia-CaCl_2_ solution was added. All samples were measured in duplicate. 

### 2.9. Lipid Peroxidation Measurement in Human Plasma

Lipid peroxidation was assessed by measuring the concentration of thiobarbituric acid-reactive substances (TBARS) in human plasma. The final concentrations of extract and fractions were 1, 5, 10, and 50 μg/mL. The method was carried out as described in Sławińska et al. [13].

### 2.10. Carbonyl Group Measurement in Human Plasma

Measuring the levels of carbonyl groups in plasma was carried out with a method involving 2,4-dinitrophenylhydrazine (DNPH). The samples were incubated with the extract or fractions at the final concentrations of 1, 5, 10, and 50 μg/mL. The method was carried out as described in Sławińska et al. [13].

### 2.11. PBM Cells Isolation

Peripheral blood mononuclear (PBM) cells were isolated from the leucocyte-buffy coat, which was collected from the blood of healthy, non-smoking donors from Blood Bank (Lodz, Poland) as described previously [14]. 

### 2.12. DNA Damage 

PBM cells were incubated (2 h at 37 °C) with plant extract and fractions (the concentration range was 1–50 μg/mL). After treatment, cells were washed and suspended in RPMI 1640 medium. Then, 25 μM H_2_O_2_ was added. PBM cells were incubated on ice for 15 min. The DNA damage was studied with the alkaline comet assay, according to Singh et al. [15], as described previously by Tokarz et al. [16]. 

### 2.13. PBM Cells Viability 

Plant extract and fractions were added to wells to obtain final concentrations of 1, 5, 10, and 50 μg/mL and incubated for 2 h and 24 h (37 °C, 5% CO_2_). The cell viability resazurin assay was performed similarly to the method described by O’Brien et al. [17]. 

### 2.14. Statistical Analysis

Results were presented as mean ± SD. For the data that did not have normal distribution, the Mann–Whitney test was used. Student’s *t*-test or ANOVA was used for the data with normal distribution. The differences were considered statistically significant at *p* < 0.05.

## 3. Results

Purification and fractionation of the extract isolated from *G. hederacea* resulted in three fractions: 20% fraction, 60% fraction, and 85% fraction. Major components of the above-mentioned preparations were tentatively identified and classified on the basis of their MS and UV spectra, chemical analysis, and literature data [18,19,20,21,22,23,24,25,26,27,28,29,30,31,32,33,34,35,36,37,38,39,40,41,42,43,44,45,46,47,48,49] (Table 1). For example, the total concentration of phenolic acids in the extract was 177.64 mg/g, while the total flavonoid content was 115.8 mg/g (Table 2 and Table 3). The main identified phenolic acids are rosmarinic acid, rosmarinic acid methyl ester, chlorogenic acid, neochlorogenic acid and among the flavonoids there are rutin, quercetin 3-[6″-(3-hydroxy-3-methylglutaryl)-galactoside] and apigenin 7-(6″-malonylglucoside). The 20% fraction consists almost exclusively of phenolic acids, of which neochlorogenic acid, 2-O-caffeoylthreonic acid, is the most abundant. No flavonoids were identified in this fraction. In turn, the 60% fraction contains phenolic acids (mostly rosmarinic acid, rosmarinic acid methyl ester and chlorogenic acid) as well as flavonoids (rutin, quercetin 3-[6″-(3-hydroxy-3-methylglutaryl)-galactoside] and apigenin 7-(6″-malonylglucoside)). In the 85% fraction, there are mainly flavonoids with the highest content of apigenin (Table 1, Table 2 and Table 3).

The analysis of coagulation times in human plasma showed that the tested plant preparations (extract and three fractions isolated from *G. hederacea* L.; concentration range: 1–50 μg/mL; incubation time: 30 min) did not affect APTT, PT, or TT (Figure 1A–C).

The extract and 85% fraction at the highest used concentration—50 μg/mL—inhibited plasma lipid peroxidation induced by H_2_O_2_/Fe^2+^ (Figure 2A). As demonstrated in Figure 2B, extract and two fractions (20 and 60% fraction; at the highest used concentration—50 μg/mL) reduced plasma protein carbonylation stimulated by H_2_O_2_/Fe^2+^. In addition, 85% fraction (at three concentrations: 5, 10, and 50 μg/mL) also inhibited protein carbonylation. At the highest concentration (50 μg/mL), protein carbonylation was decreased by approximately 50% compared to human plasma treated with only H_2_O_2_/Fe^2+^ (Figure 2B).

We have shown that the extract and fractions isolated from *G. hederacea* L. do not induce DNA damage with the exception of 60% fraction at 50 μg/mL (*p* < 0.05) (Figure 3C). All fractions and the extract were very effective at decreasing H_2_O_2_-induced DNA oxidative damage in PBM cells (Figure 3A–D). Our research has shown that the 85% fraction has the strongest protective properties against DNA damage induced by H_2_O_2_ (Figure 3D). With the exception of the sample pre-incubated with the 85% fraction at a concentration of 10 μg/mL (*p* < 0.05), all other samples showed a decrease in DNA damage to the level visible in the negative control (control (−)). Figure 4 shows representative photos of the comets, which were obtained after pre-incubation of PBM cells with extract and fractions isolated from *G. hederacea* at 50 μg/mL and followed by incubation with H_2_O_2_ at 25 μM. In microscopic slides obtained from the cells that were pre-incubated with the extract and fractions, comets with smaller tails are visible compared to comets obtained from cells incubated only with H_2_O_2_.

In our study, cell viability was measured with the resazurin reduction assay, which is based on the ability of viable cells to reduce resazurin to fluorescent resorufin [17]. We observed that the cell viability was decreased after 2 h incubation of PBM cells with extract and fractions isolated from *G. hederacea* up to 80% (Figure 5A). A decrease in cell viability at all used concentrations (1–50 μg/mL) was noted after incubation with the 20% and 85% fractions. The extract was the least cytotoxic because the decrease in viability of PBM cells was visible only after incubation with the two highest concentrations of 10 μg/mL (*p* < 0.01) and 50 μg/mL (*p* < 0.05) (Figure 5A). During the 24 h incubation of PBM cells, their viability was decreased to a maximum of about 65% (*p* < 0.001) in all used concentrations of extract and fractions isolated from *G. hederacea* (Figure 5B). 

## 4. Discussion

Various studies examined the effect of different *G. hederacea* preparations on selected biological processes both in vivo and in vitro. For example, Wang et al. [1] observed the beneficial effects of daily *G. hederacea* extracts (saline and hot water extract) supplementation against cholestatic liver injury in Sprague–Dawley rats. These effects were associated with anti-fibrotic, anti-inflammatory, and antioxidant activity. Recently, Xiao et al. [50] have used *G. hederacea* extract (Hitechol^®^), which contains saponins, essential oil, and phenolic compounds (chlorogenic acid, caffeic acid, flavonoids, and tannins), and studied its effect on gallstone formation. Normal and C57BL/6 mice with or without cholesterol gallstone were supplemented with the extract. Authors observed its beneficial effect against gallstone, which was mediated via its antioxidant properties. To evaluate its antioxidant potential, the activity of catalase and superoxide dismutase as well as the level of reduced glutathione were measured in plasma and prepared liver homogenate.

An important, novel aspect of our findings is that the tested extract and all tested fractions from aerial parts of *G. hederacea* protected human plasma proteins from damage induced by hydroxyl radicals. Moreover, the 85% fraction (a flavonoid fraction containing large quantities of apigenin) had the strongest antioxidant activity. On the other hand, all the obtained results suggest that the tested extract and three fractions did not influence in vitro coagulation in human plasma. 

The bioavailability and toxicity of phenolic compounds is an important element in the evaluation of their biological activity. For example, the research of Chao et al. [4] indicates that a hot water extract of *G. hederacea* (12.5, 25, and 50 μg/mL) ameliorates H_2_O_2_-mediated cytotoxicity and DNA damage, inhibits caspase-3 activity and apoptosis, stabilizes mitochondrial transmembrane potential, and reduces ROS production in rat pheochromocytoma line 12 (PC12) cells. The authors reported that chlorogenic acid, rutin, rosmarinic acid, caffeic acid, ferulic acid, and genistein are the most abundant phytochemicals detected in the extract. In another in vitro model, the same authors [3] investigated the cytotoxic effects of ethyl acetate fraction extract of *G. hederacea* (200–400 μg/mL) on HepG2 cells. Rosmaric acid, caffeic acid, and ferulic acid were the most abundant phenolic compounds. The authors suggest that this extract can inhibit the proliferation of HepG2 cells through intracellular ROS-mediated apoptosis.

Grabowska et al. [5] have studied cytotoxic properties of water and ethanol extracts from dried aerial parts of *G. hederacea* (10–100 μg/mL) in vitro. Cytotoxicity analysis of the extracts included two colon cancer (Caco2, HT29) and two melanoma cell lines (HTB140, A375). In addition, studies were performed on hepatoma cells HepG2, revealing the phenotype of normal hepatocytes, and normal skin keratinocytes (HaCaT) were also included. The results indicate that the extracts are not toxic to normal human cells (measured by the MTT assay) and cancer cells. Moreover, the tested extracts had good antioxidant properties which were correlated with their chemical content. The water extracts showed significantly higher antioxidant activity compared to the ethanol extracts prepared by the same method. The HPLC method was applied to determine and compare the content of phenolic acids (rosmarinic, chlorogenic, protocatechuic) and flavonoids (rutin, isoquercetin) in the extracts. HPLC analysis indicated that among phenolic acids, rosmarinic acid was the main one, with its highest content (4.28–4.89 mg/g dry plant material) in water extracts prepared by the I/ME method (infusion combined with maceration). The level of this acid was significantly lower in ethanol extracts prepared by the same method (1.07–1.14 mg/g dry plant material). Similarly to rosmarinic acid, the highest levels of chlorogenic acid were found in the water extracts prepared by the I/ME method (3.41–3.70 mg/g of dry plant material). The highest content of rutin and isoquercetin was found in ethanol extracts prepared by the HRE method (heat reflux extraction) (0.84–0.99 and 0.82–0.96 mg/g of dry plant material, respectively) [5].

Recently, Kim et al. [51] have isolated different terpenoids from *G. hederacea*, which had various biological properties, including cytotoxic activity against selected human cancer cell lines such as malignant ovarian ascites (SK-OV-3) and skin melanoma (SK-MEL-2). Some of these terpenoids exhibited inhibitory effects on NO production, a significant stimulating effect on nerve growth factor (NGF) secretion in C6 glioma cells, and a neurotrophic effect [51]. 

Another novel finding of our study is that the extract and three fractions isolated from aerial parts of *G. hederacea* can reduce DNA oxidative damage induced by hydrogen peroxide in PBM cells (Figure 3A–D). The 85% fraction had the greatest protective potential (Figure 3D). Phytochemical analysis showed that this fraction contains a high concentration of apigenin (26.17 ± 1.44 mg/g), which might be responsible for its antioxidant properties (Table 3). Apigenin (4′,5,7-trihydroxyflavone) is an important flavonoid abundant in many plants, including fruits and vegetables. Parsley, chamomile, celery, spinach, artichoke, and oregano are especially rich in apigenin. For example, dried parsley contains 45.035 μg/g of this compound [52]. Numerous in vitro and in vivo studies conducted over the last few years have shown many valuable properties of apigenin, including antioxidant, antibacterial, anti-inflammatory, and anticancer properties. Flavonoids have strong antioxidant potential and regulate many cellular processes by scavenging ROS. Recent studies have shown that apigenin can effectively prevent cyclophosphamide hepatotoxicity by inhibiting inflammatory response, oxidative stress, and apoptosis [53]. The hepatoprotective potential of apigenin is associated with the upregulation of Nrf2/HO-1 signaling and enhancement of antioxidant defenses. A more detailed description of the mechanisms of antioxidant activity of apigenin can be found in a review by Kashyap et al. [54]. 

Another flavonoid that we have identified in aerial parts of *G. hederacea* is rutin. Rutin is particularly abundant in the 60% fraction (62.31 ± 0.98 mg/g) (Table 3). The name rutin originates from *Ruta graveolens* L., which is a plant that is rich in rutin. Rutin has also been named rutoside, vitamin P, quercetin-3-O-rutinoside, and sophorin. The natural sources of rutin are fruits, medicinal herbs, and plants [55]. Numerous studies have indicated many pharmacological properties of rutin such as its antiprotozoal, antibacterial, anti-inflammatory, antitumor, antiviral, antiallergic, vasoactive, cytoprotective, antispasmodic, hypolipidemic, antihypertensive, and antiplatelet properties. It has shown huge anticancer potential against a range of cancer cell lines including glioblastoma, breast cancer, lung adenocarcinoma, prostate cancer, cervical cancer, gastric cancer, leukemia, hepatocellular carcinoma, and colon cancer cell lines [55]. Many studies have also shown the protective properties of rutin in various types of cells as well as in vivo [56,57,58]. For example, it may protect endothelial dysfunction through inhibiting Nox4-responsive oxidative stress and ROS-sensitive NLRP3 signaling pathway under high glucose stress both in vivo and vitro [56]. It was also shown that pre-treatment with rutin ameliorated the toxic effect of t-BHP by modulating the basal level of glutathione, carbonyl, and thiol groups. It also protected erythrocytes against the t-BHP-induced oxidative stress as evidenced by the augmented activity of antioxidant enzymes such as catalase, dismutase and others. The qPCR analyses showed that t-BHP potently upregulates the *iNOS* and downregulates the *Nrf2* expression, which was ameliorated with rutin treatment in a dose-dependent manner like silymarin [57].

The studies we carried out showed the presence of phenolic acids in the aerial parts of *G. hederacea*. Two of them, present mainly in the fraction 60%, are noteworthy. There are rosmarinic acid (101.52 ± 5.76 mg/g) and chlorogenic acid (31.77 ± 4.03 mg/g) (Table 2). Rosmarinic acid (O-caffeoyl-3,4-dihydroxyphenyl lactic acid) is a naturally occurring polyphenolic compound, which is abundantly distributed in herbs, such as rosemary, sweet basil and perilla [59]. It was shown that Cr-induced preneoplastic lesions on the liver and kidney tissues of rats were alleviated by rosmarinic acid through the upregulation of the Nrf2 pathway and its powerful antioxidant effects [60]. Moreover, rosmarinic acid and its derivatives can protect cells against H_2_O_2_-induced DNA damage and apoptosis [61] and UVB-induced DNA damage and oxidative stress in HaCaT keratinocytes [62].

## 5. Conclusions

This study provides information about the chemical content and biological activity of various preparations (crude extract and three fractions) from aerial parts of *G. hederacea*. Our results indicate that the 85% fraction (rich in flavonoids, mostly apigenin) has especially potent activity and could be used as valuable source of antioxidants. However, the mechanism of their antioxidant properties remains unclear and requires further studies.

## Figures and Tables

**Figure 1 nutrients-15-01671-f001:**
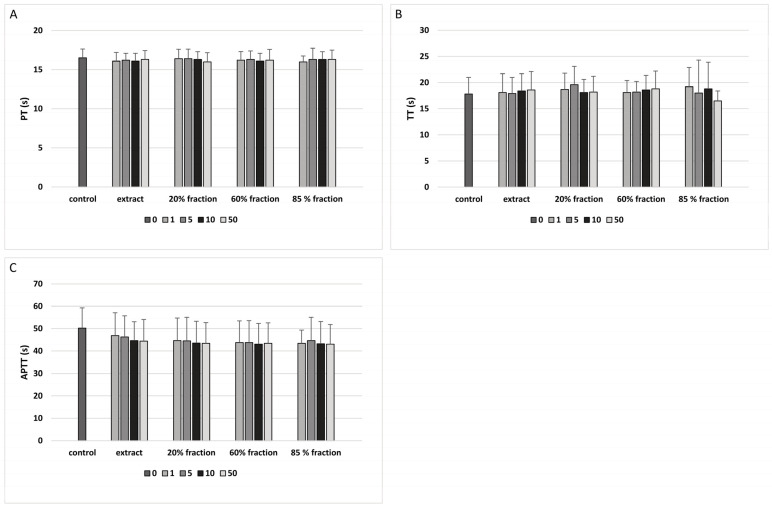
The effect of the extract and three fractions (20%, 60% and 85%; concentration range: 1–50 μg/mL; incubation time: 30 min) on the hemostatic parameters of human plasma: PT (**A**), TT (**B**), and APTT (**C**). Data represent means ± SD of 6 experiments; *p* > 0.05 (compared with control).

**Figure 2 nutrients-15-01671-f002:**
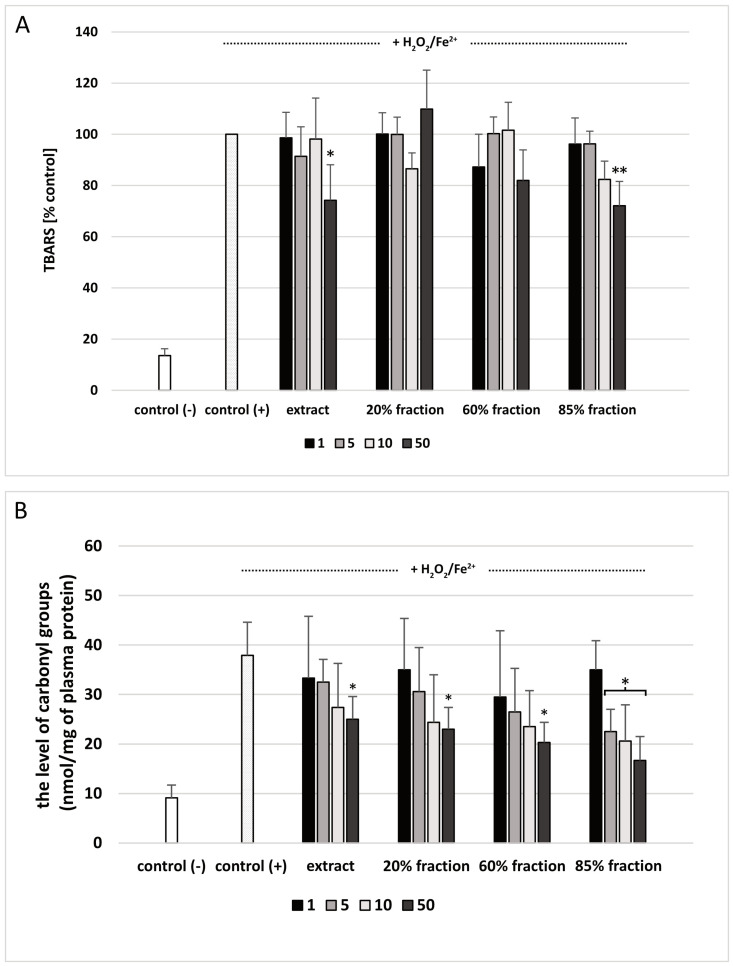
The effect of extract and three fractions (20%, 60% and 85%; concentration range: 1–50 μg/mL; incubation time: 30 min) on lipid peroxidation (**A**) and on protein carbonylation (**B**) in plasma treated with H_2_O_2_/Fe^2+^. Negative control (control (−)) refers to plasma not treated with H_2_O_2_/Fe^2+^, whereas positive control (control (+)) to plasma treated with H_2_O_2_/Fe^2+^. The differences between control (−) and control (+) were statistically significant. Data represent means ± SD of 5 experiments; * *p* < 0.05, ** *p* < 0.01, *p* > 0.05 (compared with positive control).

**Figure 3 nutrients-15-01671-f003:**
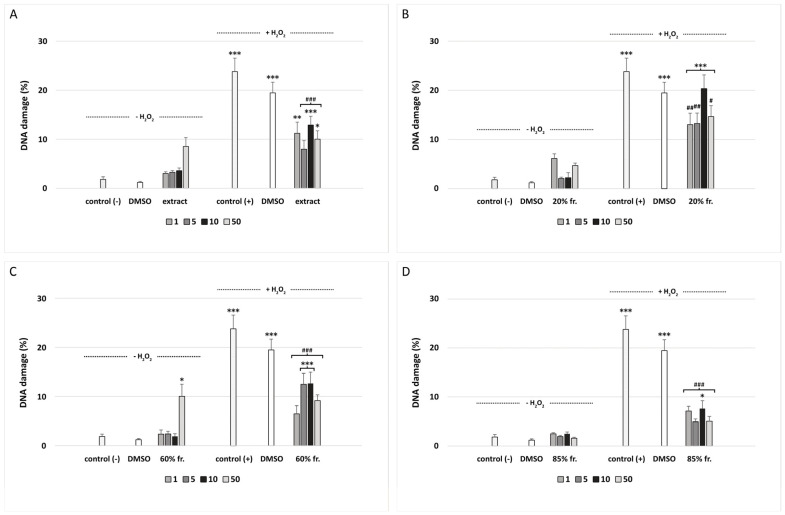
The effect of extract (**A**) and three fractions (20%, 60% and 85%; concentration range 1–50 μg/mL; pre-incubation time: 2 h) (**B**–**D**, respectively) on DNA damage in PBM cells treated with H_2_O_2_ at 25 μM for 15 min on ice. Data represent means ± SEM of 3 experiments (from different donors). * *p* < 0.05, ** *p* < 0.01, *** *p* < 0.001 (compared with control (−)); # *p* < 0.05, ## *p* < 0.01, ### *p* < 0.001 (compared with control (+)).

**Figure 4 nutrients-15-01671-f004:**
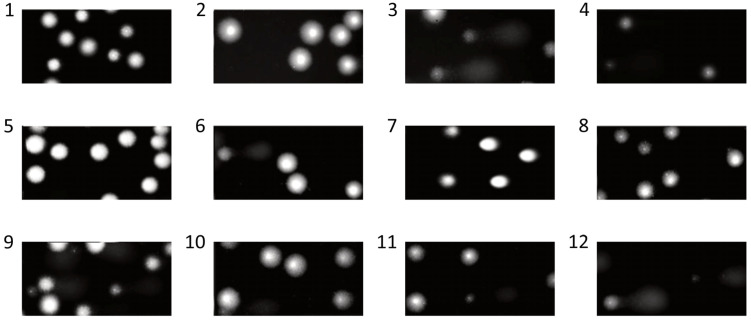
Representative photos of comets obtained in the alkaline version of the comet assay after pre-incubation of PBM cells for 2 h with extract and fractions (20%, 60% and 85%) at 50 μg/mL and incubation with 25 μM H_2_O_2_ for 15 min on ice. 1. control (−); 2. DMSO; 3. control (+) (H_2_O_2_); 4. control (+) with DMSO; 5. fraction 20%; 6. fraction 60%; 7. fraction 85%; 8. extract; 9. fraction 20% + H_2_O_2_; 10. fraction 60% + H_2_O_2_; 11. fraction 85% + H_2_O_2_; 12. extract + H_2_O_2_.

**Figure 5 nutrients-15-01671-f005:**
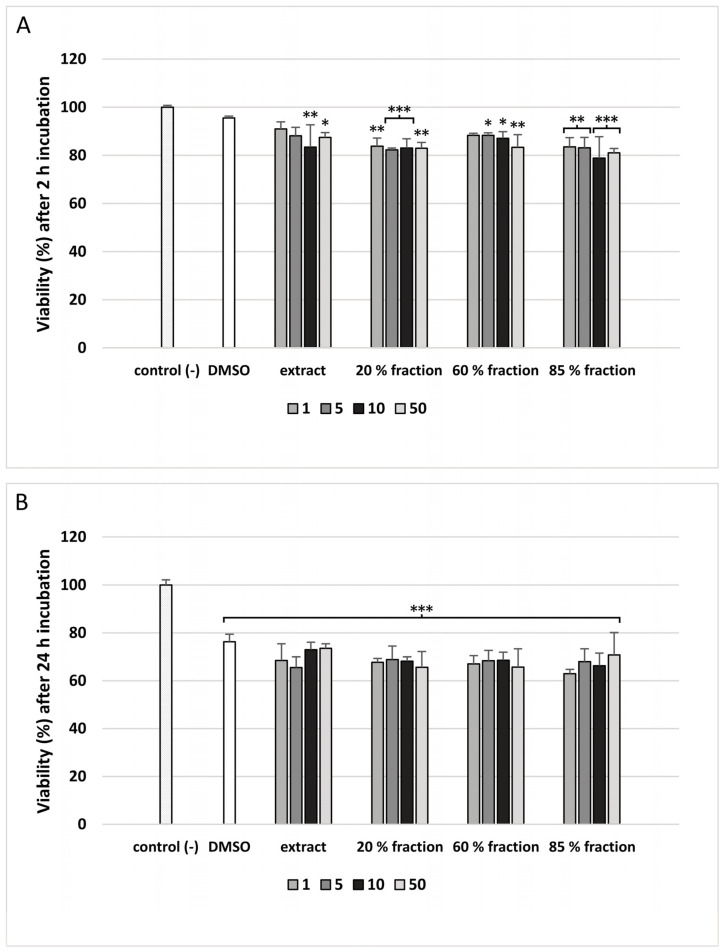
The effect of extract and fractions (20%, 60% and 85%; concentration range 1–50 μg/mL) on the viability of PBM cells. The cell viability of PBM cells was measured after 2 h (**A**) and 24 h (**B**) incubation with extract and fractions. Data represent means ± SD of 3 experiments (from different donors). * *p* < 0.05, ** *p* < 0.01, *** *p* < 0.001 compared with control (−).

**Table 1 nutrients-15-01671-t001:** Phytochemical characteristics of the crude extract and three fractions (20%, 60%, and 85%) from *G. hederacea* identified by UHPLC-QTOF MS/MS.

Peak	RT (min)	Max. *m/z* (−/+)	Ion Formula	mSigma (−)	HRMS-MS/MS Fragment (ESI−), *m*/*z*/Fragment (ESI^+^), *m*/*z*	Class	Tentative ID	References:	Area Frac. % of Extract	Area Frac. % of 20%	Area Frac. % of 60%	Area Frac. % of 85%
1	4.70	371.0632	C_15_H_15_O_11_	0.7	209.0303 (100), 191.0207 (35)	phenolic acids	4-caffeoylglucaric acid	[18]	0.46	7.61	nd	nd
2	4.81	315.0734	C_13_H_15_O_9_	5.6	315.0731 (100), 153.0185 (10.9), 109.0282 (1.4)	phenolic acids	gentisic acid 5-O-glucoside	[19]	0.23		nd	nd
3	5.21	371.0627	C_15_H_15_O_11_	4.6	209.0309 (100), 191.0201 (40.5)	phenolic acids	2-caffeoylglucaric acid	[18]	0.59	7.20	nd	nd
4	5.56	353.0889	C_16_H_17_O_9_	8.2	353.0888 (8,6), 191.0566 (100), 179.0355 (54.4), 135.0436 (7.2)	phenolic acids	3-O-caffeoylquinic acid (neochlorogenic acid)	identified by comparison to reference compound	2.75	27.34	ta	nd
5	6.11	297.0624	C_13_H_13_O_8_	3.6	179.0360 (40), 135.0289 (100)	phenolic acids	3-O-caffeoylthreonic acid	[20]	0.8	9.09	ta	nd
6	6.75	297.0622	C_13_H_13_O_8_	9.6	179.0350 (17.3), 135.0281 (100)	phenolic acids	2-O-caffeoylthreonic acid	[20]	2.77	23.98	ta	nd
7	7.24	337.0933	C_16_H_17_O_8_	11.1	191.0891 (24.6), 163.0392 (100)	phenolic acids	p-coumaroylquinic acid	[20]	0.99	9.27	ta	nd
8	7.91	353.0884	C_16_H_17_O_9_	3.4	353 (0,4, 191 (100), 179 (0,8), 173 (0,4), 161 (0,6)	phenolic acids	5-O-caffeoylquinic acid (chlorogenic acid)	identified by comparison to reference compound	4.94	20.84	5.00	nd
9	8.38	353.0887	C_16_H_17_O_9_	5.8	353.0888 (9.6), 191.0566 (100), 179.0356 (77.7), 173.0454 (72.3), 161.0246 (2.6), 135.0437 (17.3)	phenolic acids	4-O-caffeoylquinic acid (cryptochlorogenic acid)	identified by comparison to reference compound	0.99	ta	ta	nd
10	8.53	325.0572	C_14_H_13_O_9_	13.8	193.0510 (100)	phenolic acids	fertaric acid	[21]	0.38	ta	ta	nd
11	8.96	625.1414	C_27_H_29_O_17_	7.1	625.1420 (11.6), 463.0870 (100), 301.0354 (66.5)	flavonoid	quercetin-3-gentiobioside (quercetin 3-O-diglucoside)	[22]	0.79	nd	ta	nd
12	9.52	297.0620	C_13_H_13_O_8_	10.5	179.0360 (33.3), 135.0276 (100)	phenolic acids	4-O-caffeoylthreonic acid	[20]	0.98	nd	ta	nd
13	9.72	353.0885	C_16_H_17_O_9_	1.7	191.0563 (100)	phenolic acids	1-O-caffeoylquinic acid	identified by comparison to reference compound	0.48	nd	ta	nd
14	9.96	387.1662	C_18_H_27_O_9_	3.3	387.1662 (100), 207.1032 (14.3)	oxylipins	tuberonic acid glucoside	[23]	1.24	nd	ta	nd
		711.1416	C_30_H_31_O_20_	9.4	667.1517 (9.1), 505.0999 (24.1) 463.0859 (100), 301.0360 (53.6)	flavonoids	quercetin 3-O-(6″-malonylglucoside)-7-glucoside	[24,25]		nd	ta	nd
15	10.25	395.099	C_18_H_19_O_10_	5.5	335.0780 (3.5), 233.0674 (100)	phenolic glycosides	7-β-galactopyranosyl-oxycoumarin-4-acetic acid methyl ester	[26]	1.1	nd	ta	nd
16	11.98	741.1870	C_32_H_37_O_20_	14.2	741.1904 (13.7), 591.1338 (2.4), 475.0886 (1.8), 3010.0282 (100)	flavonoids	quercetin rutinoside pentoside	[27,28]	2.25	nd	3.07	nd
		755.2050	C_33_H_39_O_20_	7.2	755.2046 (12.7), 300.0279(100)	flavonoids	quercetin-3-O-(2″rhamnosyl)-7-O-rutinoside (manghaslin)	[29]		nd		nd
17	13.43	609.1460	C_27_H_29_O_16_	5.1	609.1463 (27.9), 301.0343 (100)	flavonoids	rutin (quercetin rutinoside, quercetin-3-O-α-L-rhamnopyranosyl-(1→6)-β-D-glucopyranose)	identified by comparison to reference compound	8.31	nd	11.29	nd
18	13.69	389.1822	C_18_H_29_O_9_	10.6	389.1823 (100), 227.1298 (7.6) 209.1187 (9.6)	fatty acyls	(−)-11-hydroxy-9,10-dihydrojasmonic acid 11-β-D-glucoside	[30,31]	1.74	nd	2.3	nd
19	13.82	753.1892	C_33_H_37_O_20_	21.9/12.4	609.1473 (16.1), 301.0279 (100)	flavonoids	quercetin deoxyhexsoside hexsoside 3-hydroxyl-3-methyloglutaryl	not found	3.11	nd	3.74	nd
		463.0885	C_21_H_19_O_12_	30.5/7.9	463.0899 (9.3), 301.0348 (100)	flavonoids	hyperoside (quercetin 3-galactoside)	identified by comparison to reference compound		nd		nd
20	14.05	447.0936	C_21_H_19_O_11_	4.4	447.0932 (53.1), 285.0405 (100)	flavonoids	luteolin 7-O-glucoside	identified by comparison to reference compound [32]	0.88	nd	ta	nd
21	14.24	279.1238	C_15_H_19_O_5_	5.8	279.1229 (73.8), 217.1230 (100), 165.0898 (53.7)	sesquiterpenes	phaseic acid	[33]	1.2	nd	ta	nd
22	14.59	549.0889	C_24_H_21_O_15_	7.4	505.0995 (9.5), 301.0277 (100)	flavonoids	quercetin 3-O-(6″-malonylglucoside)/quercetin 3-O-(6′′-malonylgalactoside	[34]	1.45	nd	1.75	nd
23	14.69	607.1306	C_27_H_27_O_16_	4.5	463.0887 (27.1), 301.0277 (100)	flavonoids	quercetin 3-[6″-(3-hydroxy-3-methylglutaryl)-galactoside]	[29,35,36]	3.42	nd	2.51	nd
		593.1508	C_27_H_29_O_15_	15.8	593.1509 (26.5), 285.0399 (100)	flavonoids	kaempferol 3-robinobioside	[37]		nd	2.37	nd
24	14.81	717.1457/719.1591	C_36_H_29_O_16_	7.4	(+) 521.1081 (2), 295.0599 (100), 181.0493 (8.3)	phenolic acids	yunnaneic acid G/salvianolic acid E/salvianolic acid L/isosalvianolic acid B/lithospermic acid B/clinopodic acid I	[38]	3.76	nd	4.92	nd
25	15.04	515.1199	C_25_H_23_O_12_	16.3	353.0880 (50.9), 191.0563 (100), 179.0348 (58.5)	phenolic acids	3,5-dicaffeoylquinic acid	identified by comparison to reference compound and [39]	2.08	nd		nd
26	15.11	719.1622	C_36_H_31_O_16_	3.2	359.0779 (53.2), 243.0299 (36.7), 229.0142 (49.3), 197.0459 (82.7), 179.0351 (25.2), 161.0237 (100), 135.0433 (6.1)	cyclobutane lignans	sagerinic acid	[40,41,44]	1.80	nd	4.83	nd
27	15.55	431.0985	C_21_H_19_O_10_	2.2	431.0985 (100), 269.0447 (50)	flavonoids	apigenin 7-O-glucoside	identified by comparison to reference compound and [41]	1.81	nd		nd
28	15.90	533.0946	C_24_H_21_O_14_	14.8	489.1047 (100), 285.0407 (87.9)	flavonoids	luteolin 7-O-(6″-malonylglucoside)	[42]	3.06	nd	1.63	nd
29	16.02	359.0777	C18H15O8	8.0	197.0459 (100), 179.0352 (32.1), 161.0238 (88.9), 135.0435 (5.0), 133.0275 (6.3)	phenolic acids	rosmarinic acid	identified by comparison to reference compound	15.94	nd	22.00	nd
		591.1367	C_27_H_27_O_15_	14.3	489.1041 (77.3), 447.0948 (34.3), 285.0404 (100)	flavonoids	kaempferol 3-[6″-(3-hydroxy-3-methylglutaryl)-glucoside]	[35]		nd	2.15	nd
30	17.19	717.1467	C_36_H_29_O_16_	6.7	519.0942 (3.4), 339.0515 (47.9), 321.0408 (100)	phenolic acids	yunnaneic acid G/salvianolic acid E/salvianolic acid L/isosalvianolic acid B/lithospermic acid B/clinopodic acid I	[38]	5.87	nd	8.37	nd
31	17.42	517.0991/519.1129	C_24_H_21_O_13_	14.5	473.1103 (5.5), 269.0457 (100)/519.1130 (89.9), 433.1128 (8.6), 271.0599 (100)	flavonoids	apigenin 7-(6″-malonylglucoside)	[43]	2.79	nd	4.82	nd
32	18.21	773.3969	C_38_H_61_O_16_	10.4	773.3969 (56.2), 627.3386 (100), 465.2845 (17.7)	diterpenes	diterpene dHex-Hex-HMG	not found	4.87	nd	5.82	40.37
33	18.32	367.1405	C_18_H_23_O_8_	10.8	163.0761 (100), 148.0511 (78.3)	lactones	unknown lactone	not found	1.26	nd	1.43	ta
34	18.44	373.0932	C_19_H_17_O_8_	8.9	197.0453 (95.4), 175.0404 (73.6), 135.0435 (100)	phenolic acids	3-O-methyl-rosmarinic acid	[44]	1.29	nd		ta
35	18.65	373.0934	C_19_H_17_O_8_	1.2	179.0350 (17.6), 135.0434 (100)	phenolic acids	rosmarinic acid methyl ester	[44,47]	8.58	nd	12.01	ta
		717.1469	C_36_H_29_O_16_	16.4	519.0920 (10.5), 339.0515 (100), 321.0424 (10.6), 295.0621 (5.6)	phenolic acids	yunnaneic acid G/salvianolic acid E/salvianolic acid L/isosalvianolic acid B/lithospermic acid B/clinopodic acid I	[38]		nd	nd	ta
		745.1782	C_38_H_33_O_16_	17.6	489.1197 (57.8), 445.1294 (68.1), 379.0825 (42.5), 339.0513 (100), 295.0617 (38.9), 229.0142 (22.6)	phenolic acids	dimethyl lithospermate B	[45]		nd	nd	ta
36	18.92	717.1458	C_36_H_29_O_16_	6.4	519.0925 (8.1), 339.0510 (100), 321.0430 (12.5), 295.0626 (6.4)	phenolic acids	yunnaneic acid G/salvianolic acid E/salvianolic acid L/isosalvianolic acid B/lithospermic acid B/clinopodic acid I	[38]	1.32	nd	nd	ta
37	20.43	771.3812	C_38_H_59_O_16_	6.4	591.3215 (10.6), 547.3276 (100), 465.2870 (8.3), 465.2870 (8.2), 161.0447 (46.8)	diterpens	leucasperoside C	[46]	1.67	nd	nd	26.32
38	21.59	269.0456	C_15_H_9_O_16_	4.6	269.0458 (100), 225.0562 (1.6)	flavonoids	apigenin	identified by comparison to reference compound and [49]	0.88	nd	nd	17.55
39	22.01	313.072	C_17_H_13_O_6_	5.3	161.0238 (100)	flavon	unknown flavon	not found	0.88	nd	nd	8.38
40	24.78	327.0876	C_18_H_15_O_6_	9.8	327,0875 (100), 312.0622 (32.5), 284.0656 (14.2), 242.0557 (16.6), 150.0317 (37.3)	flavon	salvigenin	[48]	0.27	nd	nd	7.37

ta—trace amounts. nd—not detected.

**Table 2 nutrients-15-01671-t002:** Content of phenolic acids in the extract from *G. hederacea* and its fractions.

Compound	Phenolic Acids (mg/g ± SD)			
	Extract	20% Fraction	60% of Fraction	85% of Fraction
4-caffeoylglucaric acid	traces	3.2 ± 0.19	ND	ND
gentisic acid 5-O-glucoside	0.82 ± 0.17	2.58 ± 0.1	ND	ND
2-caffeoylglucaric acid	1.15 ± 0.29	5.35 ± 0.71	ND	ND
3-O-caffeoylquinic acid (neochlorogenic acid)	16.6 ± 1.07	45.07 ± 1.43	1.44 ± 0.16	ND
3-O-caffeoylthreonic acid	2.69 ± 0.63	8.09 ± 0.25		ND
2-O-caffeoylthreonic acid	13.55 ± 0.06	35.35 ± 1.39	2.83 ± 0.31	Traces
p-coumaroylquinic acid	1.1 ± 0.23	3.67 ± 0.6		ND
5-O-caffeoylquinic acid (chlorogenic acid)	27.44 ± 0.54	18.58 ± 0.95	31.77 ± 4.03	ND
4-O-caffeoylquinic acid (cryptochlorogenic acid)	2.61 ± 0.64	1.7 ± 0.3	3.76 ± 0.45	ND
fertaric acid	3.32 ± 0.78	2.53 ± 0.32	4.78 ± 0.61	1.36 ± 0.08
4-O-caffeoylthreonic acid	1.87 ± 0.42	ND	2.6 ± 0.32	ND
1-O-caffeoylquinic acid	1.12 ± 0.22	ND	1.62 ± 0.22	ND
7-β-galactopyranosyl-oxycoumarin-4-acetic acid methyl ester	2.93 ± 0.60	ND	4.34 ± 0.52	ND
yunnaneic acid G/salvianolic acid E/salvianolic acid L/isosalvianolic acid B/lithospermic acid B/clinopodic acid I	5.71 ± 0.95	ND	9.14 ± 1.38	ND
rosmarinic acid	63.72 ± 1.27	ND	101.52 ± 5.76	Traces
yunnaneic acid G/salvianolic acid E/salvianolic acid L/isosalvianolic acid B/lithospermic acid B/clinopodic acid I	0.91 ± 0.55	ND	1.29 ± 0.18	ND
yunnaneic acid G/salvianolic acid E/salvianolic acid L/isosalvianolic acid B/lithospermic acid B/clinopodic acid I	7.59 ± 0.80	ND	12.3 ± 1.65	ND
3′-O-methyl-rosmarinic acid	1.46 ± 0.13	Traces	2.16 ± 0.32	ND
rosmarinic acid methyl ester	23.08 ± 0.94	Traces	36.83 ± 1.47	Traces
total phenolic acids	177.64	126.13	216.38	1.36

**Table 3 nutrients-15-01671-t003:** Content of flavonoids in the extract from *G. hederacea* and its fractions.

Compound	Flavonoids (mg/g ± SD)			
	Extract	20% Fraction	60% Fraction	85% Fraction
quercetin 3-O-(6″-malonylglucoside)-7-glucoside	2.16 ± 0.58	ND	3.85 ± 0.53	ND
quercetin-3-O-(2″rhamnosyl)-7-O-rutinoside	4.52 ± 1.19	ND	7.56 ± 0.34	ND
rutin	38.65 ± 4.92	ND	62.31 ± 0.98	Traces
hyperoside	9.03 ± 2.24	ND	14.79 ± 2.01	1.25 ± 0.1
quercetin deoxyhexsoside hexsoside 3-hydroxyl-3-methyloglutaryl; luteolin 7-O-glucoside	8.68 ± 2.28	ND	14.22 ± 2.04	ND
quercetin 3-[6′′-(3-hydroxy-3-methylglutaryl)-galactoside]	14.75 ± 4.14	ND	23.94 ± 2.89	Traces
apigenin 7-O-glucoside	6.32 ± 1.74	ND	10.14 ± 0.6	Traces
luteolin 7-O-(6″-malonylglucoside)	5.94 ± 1.99	ND	9.75 ± 0.81	ND
kaempferol 3-[6″-(3-hydroxy-3-methylglutaryl)-glucoside]	8.04 ± 2.11	ND	12.75 ± 1.8	ND
apigenin 7-(6″-malonylglucoside)	10.51 ± 2.6	ND	17.41 ± 1.65	ND
luteolin	2.05 ± 0.6	ND	3.07 ± 0.5	2.1 ± 0.19
apigenin	3.49 ± 1.03	ND	2.33 ± 0.38	26.17 ± 1.44
unknown flavon	1.66 ± 0.7	ND	1.98 ± 0.12	2.5 ± 0.24
total flavonoids	115.8		184.1	32.02

## Data Availability

Not applicable.

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
