# Peer review of "New Aspect of Composition and Biological Properties of Glechoma hederacea L. Herb: Detailed Phytochemical Analysis and Evaluation of Antioxidant, Anticoagulant Activity and Toxicity in Selected Human Cells and Plasma In Vitro"

_nutrients, 2023, doi:10.3390/nu15071671_

Round 1
Reviewer 1 Report
The manuscript “New aspect of composition and biological properties of Glechoma hederacea L. herb: detailed phytochemical analysis and evaluation of antioxidant, anticoagulant activity and toxicity in selected human cells and plasma in vitro” by Slawinska et al. presents an original research on the effect of G. hederacea phenolic compounds extract and derived fractions enriched with phenolic acids and flavonoids on selected indicators of biological activity. The manuscript is interesting and relevant to the field since the identification and evaluation of biological effects of antioxidants from various plant sources remain in scientific focus, therefore methodological as well as compositional data are of great value for further developments in the field.
However, the manuscript needs significant improvement majorly in the presentation of methodology and the discussion of the obtained results.
Specific comments for each section of the manuscript are given below.
Abstract
Please consider to shorten your abstract, it is very long.
Introduction:
1. Lines 84-99: Please move the aim of the study to a separate paragraph. The aim itself can be refined to make it more understandable. By this I mean you should remove fraction names and methodology details (eg. “H2O2/Fe2+ (the donor of the hydroxyl radical) which are more appropriate for Materials and Methods part. Focus on what was the idea of the study without going into too much detail.
Materials & Methods:
Subchapter 2.3
1. Was the medium in the ultrasonic bath thermostated at room temperature during the extraction? From my own experience, the medium in the ultrasonic bath can increase substantially when operating for longer periods of time (such as 5 h). Was the temperature controlled and how?
2. What was the sample-solvent ratio?
4. What was the volume of the extract loaded onto an RP-C18 prep column?
5. As understood from the fraction preparation of fractions, along with the 20,60 and 85% fraction, what you call “the extract” was technically again a fraction stripped from the most polar components. Is this correct? Please revise accordingly.
6. What was the elution volume for each fraction?
7. Can you please explain why did you define 85% fraction when methanol concentration in your original extract was 70%?
Subchapter 2.5
1. Line 174 and line 178: remove mentioning of the results in this part of the manuscript.
Subchapter 2.7
1. What was the number of human subjects for which the plasma was collected (how many male and female subjects) and what was their age range or average?
Results:
1. Generally, please mention all the results (all Tables) in the chapter Results.
2. Please check if you mention the right figure in the text and revise accordingly. For example, there is no Fig. 4a (line 289); Fig. 5 does not represent „comets“ (line 278).
3. Please add denotations to Fig.4 to which condition each presented „slide“ is referring.
Conclusion:
This part is very confusing. Maybe the authors have mistaken it for Discusison? However, in my opinion, the first 3 paragraphs (lines 289-335) are redundant for either discussion or conclusions. Herein, the Authors mention „common ivy“ and it remains unknown whether they mean G. hederacea „ground ivy“ of some other plant. In either way, the presented text delivers information better suited for the introduction, not for the discussion (conclusion).
Further in the text, other studies of the bioactive activity of G.hederacea extracts are presented but in a review kind of fashion. Previous studies should be presented in a way to support and broad the perspective of the results obtained in the present study. So, I advise the Authors to group the cited studies to follow the discussion of their own results. This way, they will emphasize the relevance of their results and give them more focus. As is, the reader forgets about the results of the study and gets lost in cited data, some of which I find irrelevant for this study (for example, gallstone formation effect (line 331)). I suggest the Authors refer only to studies that are closely connected to the present study.
In general, the discussion should be significantly improved and expanded.
The conclusion should be presented at the end of the manuscript and it should contain the most relevant results of the study and the importance/perspective of these findings.
The manuscript “New aspect of composition and biological properties of Glechoma hederacea L. herb: detailed phytochemical analysis and evaluation of antioxidant, anticoagulant activity and toxicity in selected human cells and plasma in vitro” by Slawinska et al. presents an original research on the effect of G. hederacea phenolic compounds extract and derived fractions enriched with phenolic acids and flavonoids on selected indicators of biological activity. The manuscript is interesting and relevant to the field since the identification and evaluation of biological effects of antioxidants from various plant sources remain in scientific focus, therefore methodological as well as compositional data are of great value for further developments in the field.
However, the manuscript needs significant improvement majorly in the presentation of methodology and the discussion of the obtained results.
Specific comments for each section of the manuscript are given below.
Abstract
Please consider to shorten your abstract, it is very long.
Introduction:
1. Lines 84-99: Please move the aim of the study to a separate paragraph. The aim itself can be refined to make it more understandable. By this I mean you should remove fraction names and methodology details (eg. “H2O2/Fe2+ (the donor of the hydroxyl radical) which are more appropriate for Materials and Methods part. Focus on what was the idea of the study without going into too much detail.
Materials & Methods:
Subchapter 2.3
1. Was the medium in the ultrasonic bath thermostated at room temperature during the extraction? From my own experience, the medium in the ultrasonic bath can increase substantially when operating for longer periods of time (such as 5 h). Was the temperature controlled and how?
2. What was the sample-solvent ratio?
4. What was the volume of the extract loaded onto an RP-C18 prep column?
5. As understood from the fraction preparation of fractions, along with the 20,60 and 85% fraction, what you call “the extract” was technically again a fraction stripped from the most polar components. Is this correct? Please revise accordingly.
6. What was the elution volume for each fraction?
7. Can you please explain why did you define 85% fraction when methanol concentration in your original extract was 70%?
Subchapter 2.5
1. Line 174 and line 178: remove mentioning of the results in this part of the manuscript.
Subchapter 2.7
1. What was the number of human subjects for which the plasma was collected (how many male and female subjects) and what was their age range or average?
Results:
1. Generally, please mention all the results (all Tables) in the chapter Results.
2. Please check if you mention the right figure in the text and revise accordingly. For example, there is no Fig. 4a (line 289); Fig. 5 does not represent „comets“ (line 278).
3. Please add denotations to Fig.4 to which condition each presented „slide“ is referring.
Conclusion:
This part is very confusing. Maybe the authors have mistaken it for Discusison? However, in my opinion, the first 3 paragraphs (lines 289-335) are redundant for either discussion or conclusions. Herein, the Authors mention „common ivy“ and it remains unknown whether they mean G. hederacea „ground ivy“ of some other plant. In either way, the presented text delivers information better suited for the introduction, not for the discussion (conclusion).
Further in the text, other studies of the bioactive activity of G.hederacea extracts are presented but in a review kind of fashion. Previous studies should be presented in a way to support and broad the perspective of the results obtained in the present study. So, I advise the Authors to group the cited studies to follow the discussion of their own results. This way, they will emphasize the relevance of their results and give them more focus. As is, the reader forgets about the results of the study and gets lost in cited data, some of which I find irrelevant for this study (for example, gallstone formation effect (line 331)). I suggest the Authors refer only to studies that are closely connected to the present study.
In general, the discussion should be significantly improved and expanded.
The conclusion should be presented at the end of the manuscript and it should contain the most relevant results of the study and the importance/perspective of these findings.
Author Response
The manuscript “New aspect of composition and biological properties of Glechoma hederacea L. herb: detailed phytochemical analysis and evaluation of antioxidant, anticoagulant activity and toxicity in selected human cells and plasma in vitro” by Slawinska et al. presents an original research on the effect of G. hederacea phenolic compounds extract and derived fractions enriched with phenolic acids and flavonoids on selected indicators of biological activity. The manuscript is interesting and relevant to the field since the identification and evaluation of biological effects of antioxidants from various plant sources remain in scientific focus, therefore methodological as well as compositional data are of great value for further developments in the field.
Response: Thank you for reviewing the manuscript and for providing such helpful comments. All of them have been taken into consideration when revising the manuscript.
However, the manuscript needs significant improvement majorly in the presentation of methodology and the discussion of the obtained results.
Specific comments for each section of the manuscript are given below.
Abstract
Please consider to shorten your abstract, it is very long.
Response: We have corrected abstract.
Introduction:
- Lines 84-99: Please move the aim of the study to a separate paragraph. The aim itself can be refined to make it more understandable. By this I mean you should remove fraction names and methodology details (eg. “H2O2/Fe2+ (the donor of the hydroxyl radical) which are more appropriate for Materials and Methods part. Focus on what was the idea of the study without going into too much detail.
Response: We have corrected this paragraph. Now, it is: “However, the effect of extracts and phenolics from G. hederacea on biomarkers of these processes is not well documented. The aim of our study was to investigate the protective effects of one extract and three frac-tions (20, 60, and 85% fraction) from G. hederacea on oxidative stress and hemostasis in vitro. We measured the levels of two biomarkers of oxidative stress in human plasma treated with H2O2/Fe2+: protein carbonylation and lipid peroxidation. Moreover, we studied the effect of this extract and these fractions on the viability of peripheral blood mononuclear (PBM) cells and the level of DNA oxidative damage induced by hydrogen peroxide. We also measured their effects on three hemostatic parameters of human plasma: prothrombin time (PT), thrombin time (TT), and activated partial thromboplastin time (APTT).”
Materials & Methods:
Subchapter 2.3
- Was the medium in the ultrasonic bath thermostated at room temperature during the extraction? From my own experience, the medium in the ultrasonic bath can increase substantially when operating for longer periods of time (such as 5 h). Was the temperature controlled and how?
Response: You are indeed right, it is not specified in the text. The extraction lasted 5 hours, but the ultrasonic bath was turned on for 15 minutes every hour, not continuously.
- What was the sample-solvent ratio?
Response: We have added: “The ratio of the sample and solvent was 1:20”.
- What was the volume of the extract loaded onto an RP-C18 prep column?
Response: We have added: “The volume of the extract loaded on the column was 5%”.
- As understood from the fraction preparation of fractions, along with the 20,60 and 85% fraction, what you call “the extract” was technically again a fraction stripped from the most polar components. Is this correct? Please revise accordingly.
Response: What we call an extract is a purified extract (without polar compounds - washing with 1% methanol, and chlorophyll - washing with 85% methanol, because 100% methanol would also wash off chlorophyll).
- What was the elution volume for each fraction?
Response: We have added: “The elution volume for each fraction was 250 mL.”
- Can you please explain why did you define 85% fraction when methanol concentration in your original extract was 70%?
Response: The extraction was performed with 70% methanol, but the crude extract was purified using Cosmosil C18-PREP to remove sugar components (eluted with 1% methanol) and chlorophyll (eluted with 85% methanol). If we used 70% methanol to wash the extract from the bed, we would not wash all the active compounds absorbed on the bed. We use the highest possible concentration of the eluent so as not to wash the chlorophyll out of the bed together with the compounds.
Subchapter 2.5
- Line 174 and line 178: remove mentioning of the results in this part of the manuscript.
Response: We have removed this part of the manuscript.
Subchapter 2.7
- What was the number of human subjects for which the plasma was collected (how many male and female subjects) and what was their age range or average?
Response: We have added this information: “Human whole blood was collected from healthy (n=6, aged 25-28) medication-free and non-smoking donors (male (n=3) and female (n=3)).”
Results:
- Generally, please mention all the results (all Tables) in the chapter Results.
Response: We have corrected this.
- Please check if you mention the right figure in the text and revise accordingly. For example, there is no Fig. 4a (line 289); Fig. 5 does not represent „comets“ (line 278).
Response: We have corrected.
- Please add denotations to Fig.4 to which condition each presented „slide“ is referring.
Response: We have added labels to each photo in Fig. 4. We have also corrected the caption for this figure.
Conclusion:
This part is very confusing. Maybe the authors have mistaken it for Discusison? However, in my opinion, the first 3 paragraphs (lines 289-335) are redundant for either discussion or conclusions. Herein, the Authors mention „common ivy“ and it remains unknown whether they mean G. hederacea „ground ivy“ of some other plant. In either way, the presented text delivers information better suited for the introduction, not for the discussion (conclusion).
Response: We have corrected this chapter. Now, it is chapter of Discussion.
Further in the text, other studies of the bioactive activity of G.hederacea extracts are presented but in a review kind of fashion. Previous studies should be presented in a way to support and broad the perspective of the results obtained in the present study. So, I advise the Authors to group the cited studies to follow the discussion of their own results. This way, they will emphasize the relevance of their results and give them more focus. As is, the reader forgets about the results of the study and gets lost in cited data, some of which I find irrelevant for this study (for example, gallstone formation effect (line 331)). I suggest the Authors refer only to studies that are closely connected to the present study.
Response: We tried to improve the discussion. Some information not directly related to our research has been removed. We have also added more detailed data about the phenolic compounds that we have identified in the fractions isolated from ivy.
In general, the discussion should be significantly improved and expanded.
Response: As above.
The conclusion should be presented at the end of the manuscript and it should contain the most relevant results of the study and the importance/perspective of these findings.
Response: We have added the conclusion.
Reviewer 2 Report
This research was aimed to investigate the phenolic compositions and protective effects of the phenolic extracts from G. hederacea L. on oxidative stress and hemostasis in vitro. The work is interesting. However, the writing is disorder, and some methods and results expression are not clear, questions and suggestions are as following.
1. the abstract is too long, some unessential research background information should be removed, and the research content and results should be summarized.
2. Phenolic compounds and their health benefits previously reported in Glechoma hederacea L. (Lamiaceae) should be provided in the introduction section.
3. qualitative analysis method in section 2.4 should be described in detail.
4. in section 3 results, the results are described too simply. Subsections are suggested to present the results in detail. For example, section 3.1 identification of phenolic compounds in the extract from Glechoma hederacea L.
5. section 4 is discussion, not conclusion. Please focus on your research result and discuss in detail, and remove the unessential contents.
6. tables should be standard tables with three lines.
7. the quality of figures are not good and low-pixel.
Author Response
Thank you for reviewing the manuscript and for providing such helpful comments. All of them have been taken into consideration when revising the manuscript.
This research was aimed to investigate the phenolic compositions and protective effects of the phenolic extracts from G. hederacea L. on oxidative stress and hemostasis in vitro. The work is interesting. However, the writing is disorder, and some methods and results expression are not clear, questions and suggestions are as following.
- the abstract is too long, some unessential research background information should be removed, and the research content and results should be summarized.
Response: We have corrected abstract.
- Phenolic compounds and their health benefits previously reported in Glechoma hederacea L. (Lamiaceae) should be provided in the introduction section.
Response: We have added this information: “Milovanovic et al. [49] studied antioxidant potential of G. hederacea as a food additive. The ethanol-water (8:2, v/v) and purified ethyl acetate extracts had significantly stronger antioxidant properties than other used extracts and commercial antioxidants such as tocopherol and butylohydroxyanizol mixture. Results of Chou et al. [2] also showed in vitro anti-oxidant potential of a hot water extract of G. hederacea (100-400 µg/mL), in which chlorogenic acid, rosmarinic acid, caffeic acid, genistein, rutin, and ferulic acid were the most abundant phytochemicals. It prevented LPS-induced DNA damage in RAW264.7 macrophages, decreased the level of malondialdehyde, increased the concentration of glutathione, and regulated the activity of antioxidant enzymes (catalase, glutathione peroxidase, and superoxide dismutase).”
- qualitative analysis method in section 2.4 should be described in detail.
Response: We have added:
The details of the qualitative analysis are described in the article Rolnik at al. and were as follows:
“The qualitative analysis of the extract and three fractions (20%, 60% and 85% were determined by high resolution LC-MS (HR-ESI-MS) analyses which were performed with the Thermo Ultimate 3000 RS (Thermo Fischer Scientific, Waltham, MS, USA) chromatographic system coupled with a Bruker Impact II HD (Bruker, Billerica, MA, USA) quadrupole-time of flight (Q-TOF) mass spectrometer and CAD detector (Charged Aerosol Detector).
The chromatographic separation was carried out on a Waters HSS T3 column (150 × 2.1 mm, 1.8 µm, Wexford, Ireland) at 40 °C, and the flow rate was 400 μL/min. A linear gradient used to separate analyses was as follows: from 2% acetonitrile in 0.1% formic acid to 99% acetonitrile in 0,1% formic acid over 22 min. The sample injection volume was 5.0 μL.
The compounds were analyzed based on data from UV and mass spectra. Electrospray ionization (ESI) was performed in negative and positive ion mode. The mass scan range was set from 80 to 2000 m/z. Ions source parameters; capillary voltage 3.0 kV, dry gas 6.0 L/min and dry temperature 200°C. The PDA was operated in the range of 190 – 750 nm. Data processing was performed using DataAnalysis 4.3 (Bruker Daltonik GmbH, Bremen, Germany).”
- in section 3 results, the results are described too simply. Subsections are suggested to present the results in detail. For example, section 3.1 identification of phenolic compounds in the extract from Glechoma hederacea L.
Response: We have added more information about identification of phenolic compounds (chapter of results): “Purification and fractionation of the extract isolated from G. hederacea L. resulted in three fractions: 20% fraction, 60% fraction, and 85% fraction. Major components of the above mentioned preparations were tentatively identified and classified on the basis of their MS and UV spectra, chemical analysis, and literature data [16–47] (Table 1). For example, total concentration of phenolic acids in the extract was 177.64 mg/g, while total flavonoid content was 115.8 mg/g (Table 2 and 3). The main identified phenolic acids are rosmarinic acid, rosmarinic acid methyl ester, chlorogenic acid, neochlorogenic acid and among the flavonoids rutin, quercetin 3-[6''-(3-hydroxy-3-methylglutaryl)-galactoside] and apigenin 7-( 6''-malonylglucoside). The 20% fraction consists almost exclusively of phenolic acids, of which neochlorogenic acid, 2-O-caffeoylthreonic acid, is the most abundant. No flavonoids were identified in this fraction. In turn, the 60% fraction contains phenolic acids (mostly rosmarinic acid, rosmarinic acid methyl ester and chlorogenic acid) as well as flavonoids (rutin, quercetin 3-[6''-(3-hydroxy-3-methylglutaryl)-galactoside] and apigenin 7-(6''-malonylglucoside)). In the 85% fraction, there are mainly flavonoids with the highest content of apigenin (Tab. 1-3).”
- section 4 is discussion, not conclusion. Please focus on your research result and discuss in detail, and remove the unessential contents.
Response: We have corrected. Section 4 is the chapter of Discussion. We have also added conclusion. Section 5 is conclusion.
- tables should be standard tables with three lines.
Response: We have corrected the formatting of the tables.
- the quality of figures are not good and low-pixel.
Response: We have provided figures with higher resolution.
Round 2
Reviewer 2 Report
The revised manuscript is obviously better than the previous one. However, there are still some points to be concerned.
1 the title and abstract still need to be concentrated to make them concise.
2 for the results, subsections were highly suggested to make the results be described fully and coherently.
3 section 2.5, did you use UPLC-MS or UPLC to determine the individual phenolic compounds?
4 Although conclusion was provided, it can not conclude the whole research work.
.